

# Postmortem transport in fossil and modern shelled cephalopods

Margaret M. Yacobucci

[1] Department of Geology, Bowling Green State University, Bowling Green, OH, United States of America

## ABSTRACT

The chambered shells of cephalopod mollusks, such as modern *Nautilus* and fossil ammonoids, have the potential to float after death, which could result in significant postmortem transport of shells away from living habitats. Such transport would call into question these clades' documented biogeographic distributions and therefore the many (paleo)biological interpretations based on them. It is therefore imperative to better constrain the likelihood and extent of postmortem transport in modern and fossil cephalopods. Here, I combine the results of classic experiments on postmortem buoyancy with datasets on cephalopod shell form to determine that only those shells with relatively high inflation are likely to float for a significant interval after death and therefore potentially experience postmortem transport. Most ammonoid cephalopods have shell forms making postmortem transport unlikely. Data on shell forms and geographic ranges of early Late Cretaceous cephalopod genera demonstrate that even genera with shell forms conducive to postmortem buoyancy do not, in fact, show artificially inflated biogeographic ranges relative to genera with non-buoyant morphologies. Finally, georeferenced locality data for living nautilid specimens and dead drift shells indicate that most species have relatively small geographic ranges and experience limited drift. *Nautilus pompilius* is the exception, with a broad Indo-Pacific range and drift shells found far from known living populations. Given the similarity of *N. pompilius* to other nautilids in its morphology and ecology, it seems unlikely that this species would have a significantly different postmortem fate than its close relatives. Rather, it is suggested that drift shells along the east African coast may indicate the existence of modern (or recently extirpated) living populations of nautilus in the western Indian Ocean, which has implications for the conservation of these cephalopods.

## INTRODUCTION

### The problem of postmortem transport

Biogeographic distributions strongly influence the ecology, evolution, and extinction of clades. While scientists studying modern organisms can collect data on the observed locations of sampled living specimens, paleontologists must assume that fossil localities provide an accurate estimate of the group's living geographic range. For most marine animal groups, especially benthic invertebrates, that assumption appears to work well, as their hard parts tend to be buried and fossilized at or near their life locations (*Kidwell & Flessa, 1996*;

Corresponding author
Margaret M. Yacobucci,
mmyacob@bgsu.edu

*Tomašových & Kidwell, 2009*; *Tyler & Kowalewski, 2017*). However, cephalopod mollusks with chambered shells present a special challenge—once their soft parts are removed, cephalopod shells can float after death and therefore be picked up and transported by surface currents (Fig. 1). This postmortem transport or "drift" has been observed in living nautilid species, but how commonly it occurs in ancient cephalopod groups has been debated in the scientific literature for over 100 years (*Walther, 1897*; *Reyment, 1958*; *Reyment, 1970*; *Reyment, 1973*; *Reyment, 2008*; *Stenzel, 1964*; *House, 1973*; *House, 1987*; *Chamberlain Jr & Weaver, 1978*; *Wani et al., 2005*; *Mapes et al., 2010a*; *Wani & Gupta, 2015*; *Yacobucci, 2015*). If postmortem transport was frequent and extensive, the geographic distributions of fossil cephalopods should be considered unreliable proxies for living ranges. Biogeographic studies of fossil cephalopods require some assurance that locality data are meaningful, especially as paleontologists investigate paleobiogeographic processes with more quantitative analytical methods (*Brayard, Escarguel & Bucher, 2007*; *Dera et al., 2011*; *Brayard et al., 2015*; *Ifrim, Lehmann & Ward, 2015*; *Korn & De Baets, 2015*; *Lehmann et al., 2015*; *Zacaï et al., 2016*; *Zacaï et al., 2017*; *Rojas et al., 2017*; *Wani, 2017*; *Yacobucci, 2017*). Also, as modern nautilids experience greater fishery pressure and habitat disruption due to climate change and other anthropogenic impacts, conservation efforts will need to accurately assess population distributions (*Dunstan, Alanis & Marshall, 2010*; *Dunstan, Bradshaw & Marshall, 2011*; *Dunstan, Ward & Marshall, 2011a*; *Dunstan, Ward & Marshall, 2011b*; *Sinclair et al., 2011*; *De Angelis, 2012*; *Barord et al., 2014*; *Williams et al., 2015*; *Ward, Dooley & Barord, 2016*; *Saunders, Greenfest-Allen & Ward, 2017*). It is therefore imperative to determine the likely frequency of postmortem transport in both fossil and modern shelled cephalopods.

In the 19th and early 20th centuries, some workers noted that the small geographic ranges and concentrations of undamaged fossil cephalopod shells indicated little postmortem transport, while others argued that the lack of preserved soft parts (particularly for ammonoids) indicated that the shells must have drifted for some time before sinking to the seafloor (*Walther, 1897*; *Reyment, 1958*); it should be noted that this lack of soft parts could potentially also be explained by a high level of ammonia in the soft tissues, which prevents mineralization of the decaying tissues, see (*Clements et al., 2017*). The "drift" argument was bolstered by observations in the mid-20th century of varyingly broken and encrusted shells of modern *Nautilus* on beaches and other coastal settings in Australia (*Iredale, 1944*), Japan (*Kobayashi, 1954*), Thailand (*Hamada, 1964*; *Toriyama et al., 1965*), and the Bay of Bengal (*Teichert, 1970*). These authors inferred from these observations that *Nautilus* shells were being transported by ocean currents away from living populations, likely in the Philippines, Indonesia, and New Caledonia. *Reyment (1958)* provided a review of much of this earlier literature. Reyment claimed that postmortem drift was common in both modern and fossil shelled cephalopods (which he described as the minority viewpoint at the time) and investigated the buoyancy of empty shells to determine which shell characteristics were related to greater buoyancy and therefore postmortem transport (see "Cephalopod shell buoyancy" below). Reyment continued throughout his career to argue vigorously that postmortem drift in cephalopods was "the rule rather than the exception" (*Reyment, 1970*; *Reyment, 1973*; *Reyment, 2008*) and influenced the position of many other paleontologists
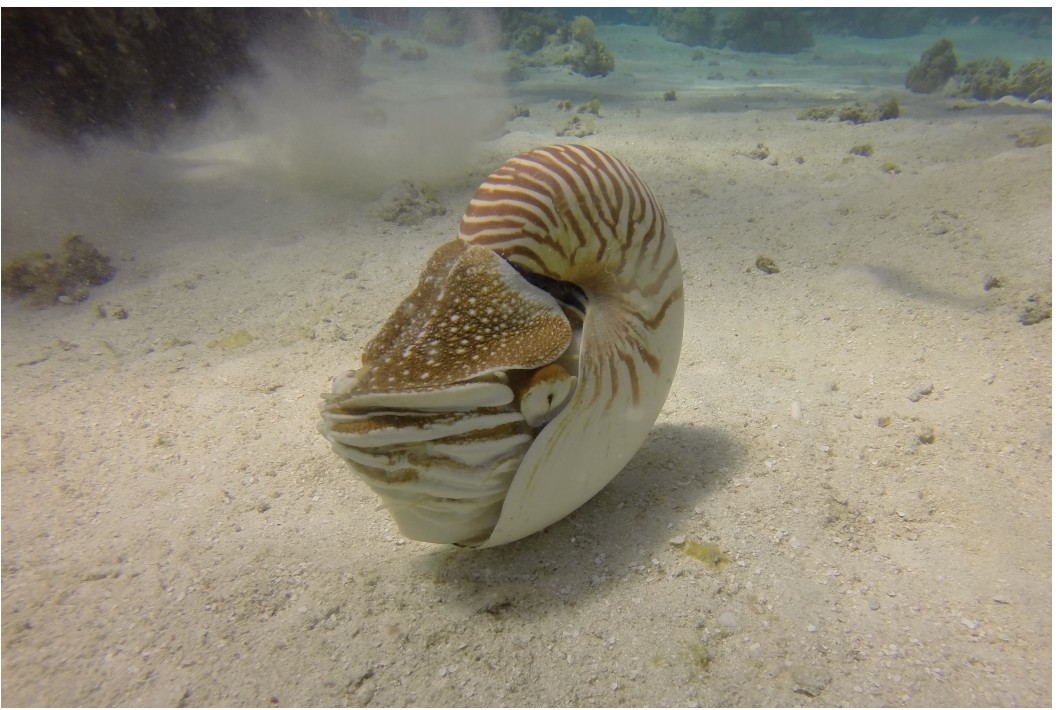

**Figure 1** **Modern *Nautilus*.** Living *Nautilus* in Palau. Photo source: image accessed from https://pxhere.com/en/photo/660195 under a Creative Commons CC0 license.

(e.g., *Stenzel, 1964*; *House, 1973*; *House, 1987*; *Cecca, 2002*) as well as biologists (e.g., *Reid, 2016*). For example, *Chamberlain Jr & Weaver (1978)* claimed that modern nautilid shells "often" rise to the surface after death and are "often dispersed by currents–sometimes on an impressive scale...thousands of miles from the habitat of the animals that build them...and similar behavior is inferred for many fossil cephalopods..." (*Chamberlain Jr & Weaver, 1978*, p. 673–674).

On the other hand, the extent to which postmortem transport has affected the distribution of fossil cephalopods has been questioned. Many fossil localities include very large numbers of well-preserved, intact shells, which is hard to explain by postmortem transport (*Kennedy & Cobban, 1976*). It is also well-established that many cephalopod taxa or morphotypes are strongly tied to particular depositional environments and sedimentary facies, implying fidelity of dead shells to their living habitat (*Tanabe, 1979*; *Lukeneder, 2015*). Paleontologists have noted that postmortem transport is likely to remove drift shells from the fossil record entirely, as they degrade over time and end up destroyed when carried by currents into high energy shoreline environments (*Hewitt, 1988*; *Maeda & Seilacher, 1996*; *Maeda, Mapes & Mapes, 2003*; *Mapes et al., 2010b*; *Hembree, Mapes & Goiran, 2014*; *Tomašových et al., 2016*; *Tomašových et al., 2017*). Fossil cephalopods, therefore, likely represent shells that sank soon after death.

*Chamberlain Jr, Ward & Weaver (1981)* used buoyancy calculations to argue that most ammonoids, especially those with shells less than 5 cm diameter or found in deeper

water settings (where ambient hydrostatic pressure is high), would quickly sink to the sea floor after the removal of soft parts, as water flooded the phragmocone (the chambered portion of the shell). A similar argument has been made for Paleozoic and Mesozoic nautiloids (*Hewitt & Westermann, 1996*; *Chirat, 2000*), although *Chirat (2000)* suggested that the Cenozoic nautilid *Aturia* (*Bronn, 1838*) experienced unusually extensive drift because its very long septal neck would have slowed the rate of flooding. *Wani et al. (2005)* conducted experiments on modern *Nautilus pompilius* (*Linnaeus, 1758*) and showed that the phragmocone floods quickly once the mantle tissue is detached, especially if the shell is small. They argued that only shells over 20 cm in diameter were likely to experience significant postmortem transport, consistent with their claim that observations of long-distance postmortem drift in modern nautilids—while sometimes dramatic—are actually quite rare.

*Mapes et al. (2010a)* pointed out that we typically only encounter modern nautilid shells that have floated into nearshore settings and have little firsthand information about what happens to nautilid shells that sink quickly to the sea floor. Remarkable data from dredged specimens of *Nautilus macromphalus* (*Sowerby, 1848*) show nautilid shells do indeed accumulate on the sea floor in deep water, near their living habitats (*Roux, 1990*; *Roux et al., 1991*; *Mapes et al., 2010a*; *Seuss et al., 2015*; *Tomašových et al., 2016*; *Tomašových et al., 2017*). Evidence of bioerosion and encrustation has been used as evidence for the duration of transport and exposure at the sea floor in both modern (*Seuss et al., 2015*; *Tomašových et al., 2016*; *Tomašových et al., 2017*) and fossil nautilids (*Luci & Cichowolski, 2014*) and in ammonoids (*Luci, Cichowolski & Aguirre-Urreta, 2016*). Such detailed taphonomic analyses are necessary, as workers have argued that one must not assume postmortem transport but rather search for evidence of it in specific cases (*Wani, 2004*; *Wani et al., 2005*; *Wani & Gupta, 2015*; *Yacobucci, 2015*).

## Cephalopod shell buoyancy

For a cephalopod shell to experience postmortem drift, it must first become positively buoyant after the death of the animal. Numerous workers have investigated the buoyancy of cephalopod shells, both during life and after death, via physical experiments and mathematical models (*Trueman, 1941*; *Denton & Gilpin-Brown, 1966*; *Westermann, 1971*; *Chamberlain Jr & Weaver, 1978*; *Ward & Martin, 1978*; *Ward & Greenwald, 1982*; *Greenwald & Ward, 1987*; *Jacobs & Chamberlain Jr, 1996*; *Kröger, 2002*; *Hammer & Bucher, 2006*; *Naglik, Rikhtegar & Klug, 2014*; *Tajika et al., 2014*; *Hoffmann et al., 2015*; *Naglik et al., 2015*), with new imaging and computational techniques (*Hoffmann & Zachow, 2011*; *Hoffmann et al., 2015*; *Lemanis et al., 2015*; *Peterman, Barton & Yacobucci, 2018*) enabling ever more sophisticated analyses. Workers have established that, in life, modern nautilids are generally neutral to slightly negatively buoyant (*Ward & Martin, 1978*; *Greenwald & Ward, 1987*). Newly formed chambers of the phragmocone are emptied of liquid slowly, at a rate of about 1 mL per chamber per day, via the siphuncle, a tube of tissue extending back from the soft body of the animal through openings in the septal walls that define the chambers (*Ward & Martin, 1978*). Modern nautilids are capable of partially refilling empty chambers in 10 to 30 h (for a rate of about 2 mL/day) in response to sudden increases in

buoyancy, for instance, if some of its shell is removed by a predator (*Ward & Greenwald, 1982*). This partial refilling, though, may be insufficient for the animal to regain neutral buoyancy.

After a shelled cephalopod dies and some or all of its soft parts are removed, and assuming it died at a shallow enough depth not to sink immediately (*Chamberlain Jr, Ward & Weaver, 1981*; *Maeda & Seilacher, 1996*), the shell will become positively buoyant and float until the phragmocone is filled by seawater moving through the siphuncular opening (assuming the phragmocone is intact) (*Wani & Gupta, 2015*). A critical question is how quickly this flooding of the phragmocone happens. *Wani et al. (2005)* conducted taphonomic experiments in which freshly killed *N. pompilius* in the Philippines were set on deep (320 m) and shallow (50 m) seafloors. Soft parts were left within the shells. They found that flooding of the phragmocone did not occur immediately postmortem, but that most shells at both depths were flooded within 3 (shallow) to 7 (deep) days, as the mantle's attachment to the last septum failed. In the few days before flooding of the phragmocone began, the dead cephalopod presumably remained neutral to slightly negatively buoyant, as in life; the dead animal would therefore not be transported by surface currents during this interval, but some transport via deeper-water currents might occur (although *Wani et al.*'s *2005* experimental design did not test for this). Flooding began when decaying mantle tissue pulled away from the back of the body chamber; presumably removal of soft parts by predation or scavenging would hasten the initiation of chamber filling. *Wani et al. (2005)* further argued that the rate of flooding would be a function of the radius, thickness, and length of the siphuncular tube. Since fossil ammonoids possessed a siphuncle 1.5 to 3 times as long as that of modern *N. pompilius*, they predicted that ammonoid phragmocones would flood faster than modern nautilids.

*Reyment (1973)* explicitly connected variations in shell form to the likelihood of positive postmortem buoyancy in ammonoids. He used modern nautilid shells and plastic models of a variety of fossil cephalopod shell morphotypes in laboratory experiments to identify how differing shell shapes produced different degrees of buoyancy in the empty shells. Plastic models were constructed based on molds of actual fossil specimens. These included three ceratite ammonoid taxa (*Ceratites* (*Acanthoceratites*) *spinosus Philippi, 1901*, *Ceratites nodosus* (*Buch, 1850*), and *Discoceratites* sp. (no author given)), which are similar in most shell parameters and ornamentation, but show a continuum of more compressed (*Discoceratites* sp.) to more inflated (*C.* (*A.*) *spinosus*) whorl shapes. Flotation experiments were conducted on these models to determine how much weight would need to be added to cause the floating empty shells to sink. *Reyment (1973)* found that the ceratite shells with more inflated whorl sections were more buoyant, that is, more weight needed to be added to the shell to counteract buoyancy forces and cause the shell to sink. The ceratite models showed that these shell shapes would have required more water in their phragmocone chambers when alive than modern *Nautilus pompilius* does, in order to maintain neutral buoyancy. *Reyment (1973)* attributed this difference to the ceratites' shells being more evolute and with a longer body chamber that *Nautilus*.

Having found experimentally that shell parameters like coiling, inflation, and body chamber length are important determinants of postmortem buoyancy, *Reyment (1973)*

argued that quantitative analysis of cephalopod shell shape would provide a more efficient way of investigating the hydrostatics of fossil shells than the time-consuming and expensive physical modeling and tank experiments he conducted. To demonstrate this, he performed a principal coordinates analysis on five shell shape parameters (shell diameter, maximum inflation, body chamber length, ventral inflation, and umbilical width) for 42 Mesozoic ammonoid species. The position of each species on a plot of the first two principal coordinates indicates its overall shell shape (Fig. 2). Based on his experimental results, Reyment then mapped out on the plot which shell shapes would have more or less buoyancy postmortem, that is, need more or less water flooding into the phragmocone to sink. He concluded that serpenticones (narrow shells with a wide umbilicus, plotting high on the second principal axis) and oxycones (disc-shaped shells with a high whorl expansion rate, plotting high on the first principal axis but low on the second principal axis) were less buoyant while the more inflated spherocone shells (plotting low on the first principal axis) showed higher postmortem buoyancy. Hence, it can be predicted that spheroconic ammonoids would be more likely to experience extensive postmortem transport.

## Linking shell form, postmortem transport potential, and geographic distribution

While Reyment himself insisted that postmortem drift was pervasive among all fossil cephalopods *(2008)*, his own work *(1973)* suggested a more complex relationship between shell form, postmortem buoyancy, and transport potential. *Reyment*'s *(1973)* morphospace for Mesozoic ammonoids (Fig. 2) closely resembles the ternary Westermann Morphospace devised by *Ritterbush & Bottjer (2012)* to capture morphological variations in regularly coiled ammonoids (Fig. 3), with three end-member forms representing serpenticones, oxycones, and spherocones. This resemblance is understandable given the overlap in shell shape parameters that the two approaches use to construct their morphospaces. Westermann Morphospace has the advantage of being expressed in a standardized frame of reference, making comparisons among any taxa possible, in contrast to more traditional factor analytic techniques that define axes that apply only to the specimens used to define them. Westermann Morphospace, then, becomes a useful framework to assess the link between shell form and postmortem transport potential.

Given the ongoing debate about the frequency of postmortem transport in both fossil and modern shelled cephalopods and the importance of this question to our understanding of the geographic distributions of these groups, the present research has three goals:

(1) To determine the relationship between cephalopod shell form and postmortem transport potential,

(2) To test the hypothesis that fossil cephalopods with higher postmortem transport potential will show artificially larger geographic ranges than cephalopods with lower transport potential,

(3) To assess the extent of postmortem transport in modern nautilid species.

Ultimately, the intention of this work is to enable fossil cephalopod workers to use shell forms to better predict the likelihood of postmortem transport and to evaluate their confidence in the use of geographic ranges derived from fossil localities as a proxy for the

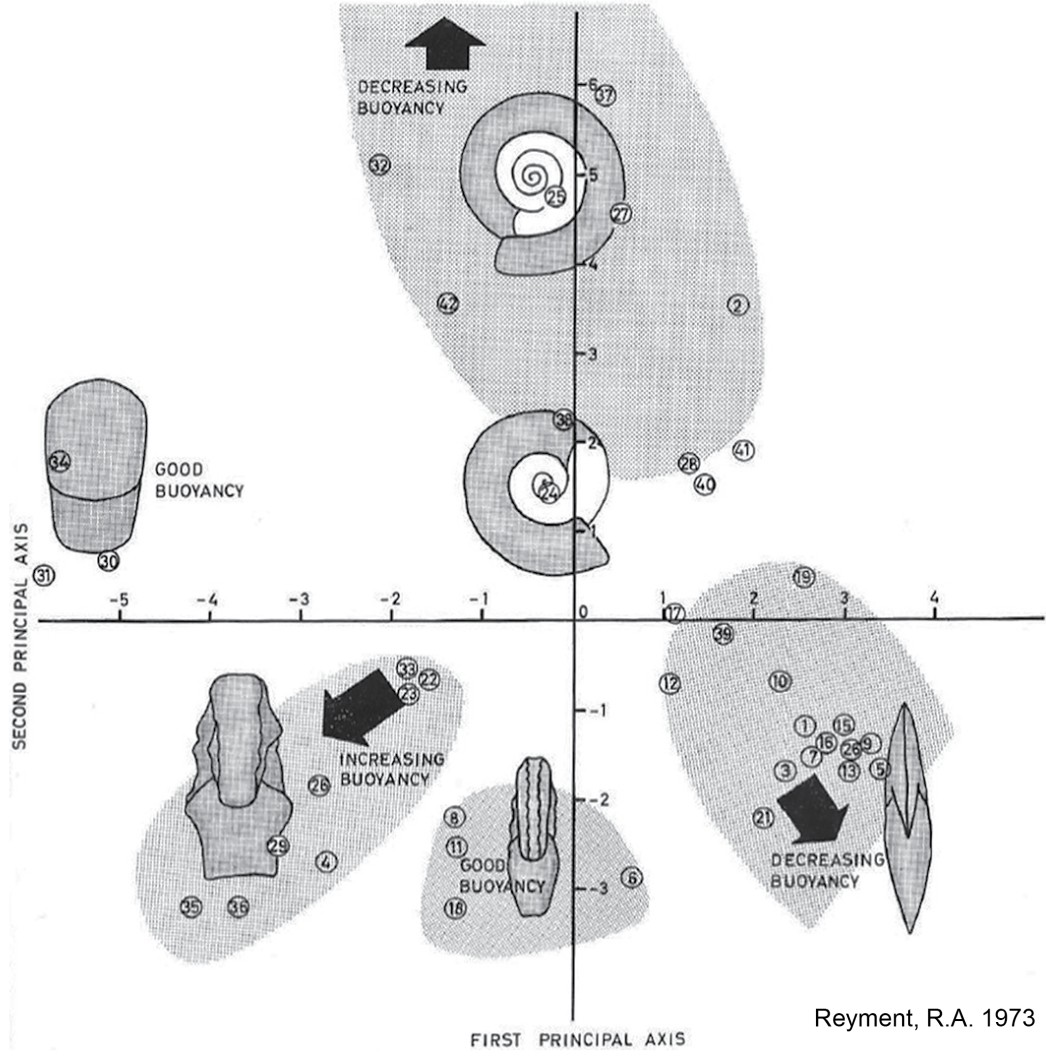

**Figure 2** **Relationship between shell form and postmortem buoyancy.** Plot of first two principal coordinates derived from *Reyment*'s *(1973)* analysis of five shell parameters (shell diameter, maximum inflation, body chamber length, ventral inflation, and umbilical width) on 42 Mesozoic ammonoid species (numbered circles). Each position on the plot reflects a particular shell shape. Reyment indicated predicted postmortem buoyancy of different shell forms on the plot, based on his experimental work that related shell shape parameters to buoyancy of the empty shell. Image source: *Reyment (1973)*, fig. 32, p. 34.

living geographic ranges of the groups under study. In addition, analysis of the geographic distribution of modern living and dead drift nautilid shells will help researchers better evaluate the geospatial extent of these charismatic but threatened species.

# MATERIALS & METHODS

## Datasets

Four datasets were compiled for morphometric and geographic analyses. All datasets are available as Supplemental Information 1.

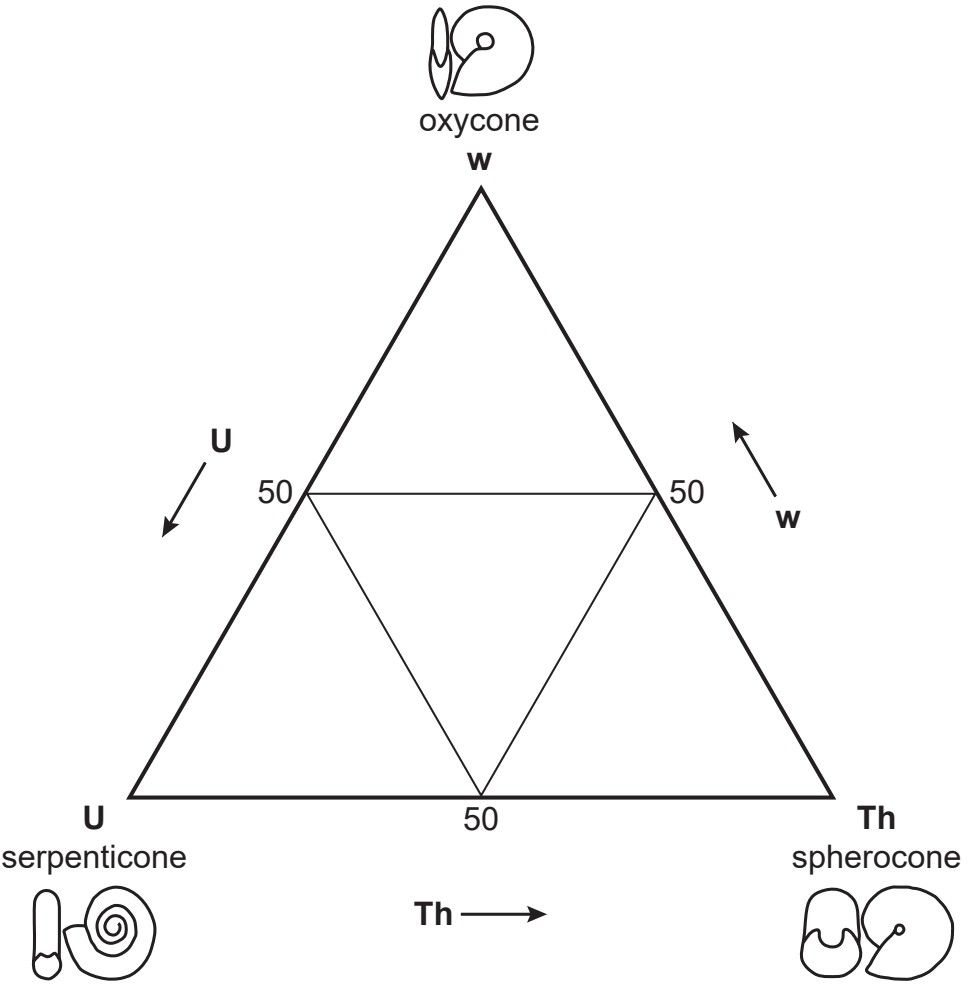

**Figure 3  Westermann Morphospace.** Schematic representation of Westermann Morphospace (*Ritterbush & Bottjer, 2012*) as a ternary diagram. Each corner of the diagram represents the maximum value for one of the three shell shape parameters used to construct the morphospace: umbilical ratio U (serpenticones), whorl thickness ratio Th (spherocones), and whorl expansion rate w (oxycones). Any planispiral shell form will plot within the ternary diagram based on its values for these three parameters. Note the similarity of Westermann morphospace to *Reyment*'s *(1973)* principal coordinates plot (Fig. 2).

(1) Shell shape data for taxa included in *Reyment (1973)* plus modern nautilids. Unfortunately, Reyment did not include the raw shell shape data upon which he performed the principal coordinates analysis shown in Fig. 2. Therefore, for each of the genera he included, images of fossil specimens were located in the ammonoid Treatise volumes (*Arkell et al., 1957*; *Wright, Callomon & Howarth, 1996*). In some cases, the images were of the same species Reyment used, while in other cases a congeneric species had to be used, based on the availability and quality of preservation of the specimens. Not having access to Reyment's specimens, the rationale for using the ammonoid Treatise volume was to use a widely available and well-known publication as a data source, making the data set more easily replicable. These images were used to measure the five shell shape

characters necessary for determining a taxon's position in Westermann Morphospace (*Ritterbush & Bottjer, 2012*): shell diameter (*D*), width of last whorl (*b*), height of last whorl (*a*), height of whorl 180° back from last whorl (*a′*), and umbilical diameter (*UD*) (Fig. 4). In total, 38 species were included in this dataset. Reyment's assessment of the degree of postmortem buoyancy for each taxon (Fig. 2) was used to assign each species to one of three categories: buoyant, intermediate, or non-buoyant. Note here that "non-buoyant" really means minimally buoyant shells that would sink with little additional weight added as water flooding the phragmocone, as opposed to "buoyant" shells that would require more weight to be added in order to sink.

For comparison, shell shape measurements were also made on examples of four living nautilid taxa: *Nautilus belauensis* (*Saunders, 1981*), *N. macromphalus* (*Sowerby, 1848*), *N. pompilius* (*Linnaeus, 1758*), and *Allonautilus scrobiculatus* (*Ward & Saunders, 1997*). Shell photographs were accessed via the website *Conchology Inc. (2017)*. These four taxa are known to sometimes drift postmortem and were therefore classified as having shell forms that are buoyant postmortem.

(2) Shell shape data from *Ritterbush & Bottjer (2012)* ammonoid dataset. In the supplementary materials for their 2012 paper, Ritterbush and Bottjer supplied the scaled and normalized Westermann Morphospace parameters *U*, *Th*, and *w* for 177 ammonoid species. These species represent a broad range of ages, clades, and shell forms. The parameters *U*, *Th*, and *w* are defined formally in the next section, "Westermann Morphospace and postmortem buoyancy".

(3) Geographic range and shell shape data for Late Cenomanian and Early Turonian (Late Cretaceous) cephalopods from *Yacobucci (2017)*. *Yacobucci (2017)* compiled geographic range data for 41 Late Cenomanian and 34 Early Turonian ammonoid genera plus the nautilid genus *Eutrephoceras Hyatt, 1894* (which was present in both substages). This dataset therefore offers the opportunity to test whether certain ammonoid shell forms are more likely to show artificially larger geographic ranges due to postmortem drift. Geographic range was estimated as the $\log_{10}$ of the area in square kilometers of a convex hull encompassing all occurrences of each taxon; see *Yacobucci (2017)* for further details. To determine the position of these genera in Westermann Morphospace, images of specimens of each genus were located in the ammonoid Treatise volumes (*Arkell et al., 1957*; *Wright, Callomon & Howarth, 1996*) and the nautiloid Treatise volume (*Teichert et al., 1964*) for *Eutrephoceras*. These images were used to measure the necessary shell shape characters.

(4) Modern nautilid occurrences from *Toriyama et al. (1965)* and *House (1987)*. *House (1987)* compiled descriptive locality information and produced a map for occurrences of both living specimens and dead drift shells of modern nautilids, based partly on occurrences reported in *Toriyama et al. (1965)*. Localities were compiled from both sources and species and place names updated as needed. Each location was then georeferenced to the nearest 0.1 decimal degrees latitude and longitude, using Google Earth. In some cases, a single numbered locality was split into multiple locations, as *House (1987)* lumped together in his locality descriptions places that are actually relatively far apart (e.g., two separate islands within one island chain). A total of 184 separate nautilid occurrences were included in the final dataset.

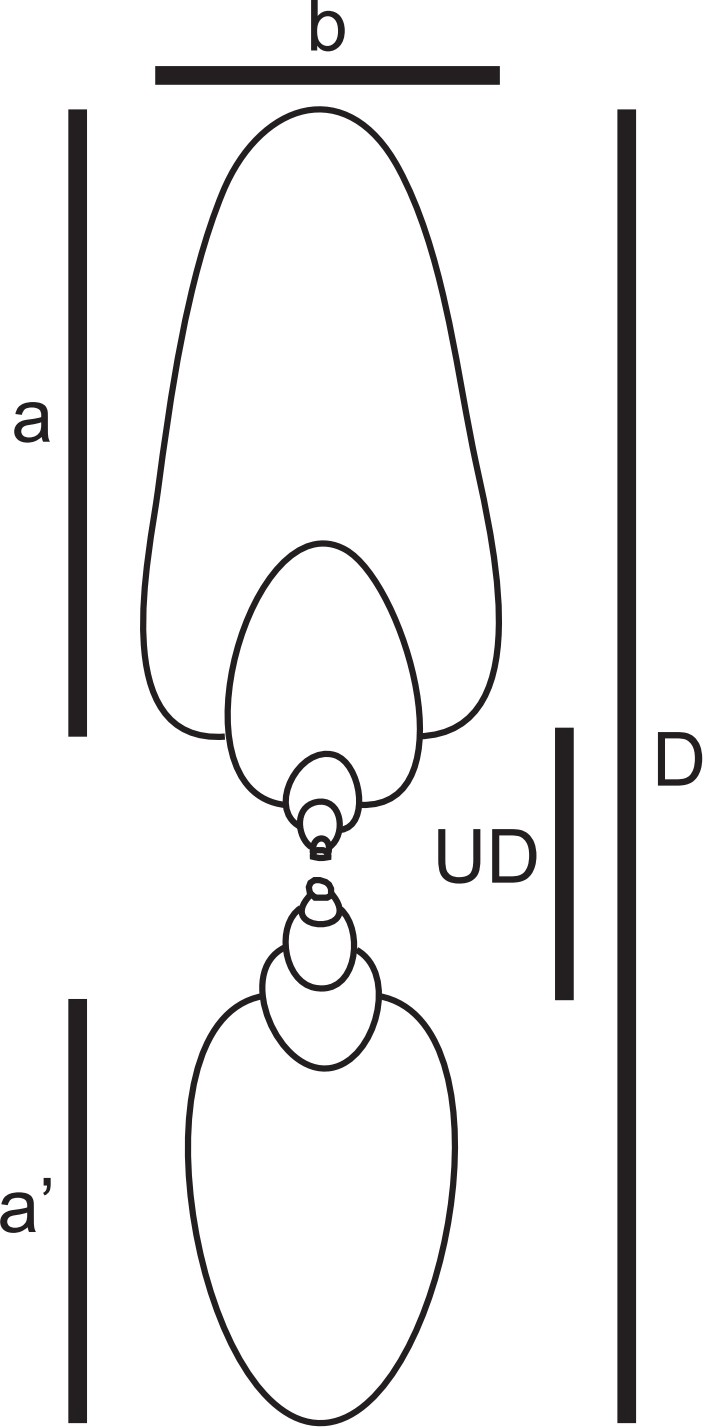

**Figure 4** **Shell shape parameters.** Diagram showing definitions of measured parameters used to characterize ammonoid shell shape. *a*, height of last whorl; *a′*, whorl height one-half-whorl back; *b*, breadth of last whorl; *D*, shell diameter; *UD*, umbilical diameter.

## Westermann Morphospace and postmortem buoyancy

Five shell shape measurements ($a$, $a'$, $b$, $D$, and $UD$; Fig. 4) were used to calculate the three parameters defining Westermann Morphospace, following the protocol of *Ritterbush & Bottjer (2012)*. First, raw values of the three key shape parameters of involution (umbilical ratio $U$), shell inflation (thickness ratio $Th$), and whorl expansion rate ($w$) were calculated:

$$\text{Raw } U = UD/D$$
$$\text{Raw } Th = b/D$$
$$\text{Raw } w = a/a'.$$

Next, these three raw parameters were scaled to fall within the range of values for common ammonoids, using the minimum and maximum values provided in *Ritterbush & Bottjer (2012*, table 2):

$$\text{Scaled } U = \text{Raw } U / 0.52$$
$$\text{Scaled } Th = (\text{Raw } Th - 0.14)/(0.68 - 0.14)$$
$$\text{Scaled } w = (\text{Raw } w - 1.00)/(1.77 - 1.00).$$

Finally, the scaled values were normalized to range from 0 to 1:

$$U = \text{Scaled } U /(\text{Scaled } U + \text{Scaled } Th + \text{Scaled } w)$$
$$Th = \text{Scaled } Th /(\text{Scaled } U + \text{Scaled } Th + \text{Scaled } w)$$
$$w = \text{Scaled } w /(\text{Scaled } U + \text{Scaled } Th + \text{Scaled } w).$$

These normalized parameters were used to construct ternary plots of Westermann Morphospace. Each corner of the ternary plot represents the maximum end member value for one of the three shell shape parameters. Individual shell forms plot within the triangular morphospace based on the values of these three parameters.

Westermann Morphospace provides a standardized framework for comparing the morphology of any normally coiled cephalopods in a reproducible way, unlike sample-specific approaches to visualizing shape variation, such as principal components analysis (PCA), which produces morphospace projections that change depending on the specimens included in the analysis. Occupation patterns within Westermann Morphospace can also be interpreted in terms of mode of life (e.g., nektonic, planktonic, demersal, or vertical migrant), a feature lacking in other morphospace approaches. On the other hand, Westermann Morphospace is based on scaled, normalized parameters, several computational steps removed from the original measurement data. To confirm that the data processing involved in Westermann Morphospace construction did not affect the general pattern observed, the five measured shell shape characters ($a$, $a'$, $b$, $D$, and $UD$) were also subjected to principal components analysis (PCA) on their correlation matrix, as well as a multi-group discriminant analysis (canonical variates analysis; CVA) with groups defined as buoyant postmortem, not buoyant postmortem, and intermediate in buoyancy, based on *Reyment*'s *(1973)* results and normalized shell thickness ratio $Th$ (see Results below, Fig. 5). PCA and CVA were conducted in PAST 3.20 (*Hammer, Harper & Ryan, 2001*); data sets and detailed results are provided in Supplemental Information 1.
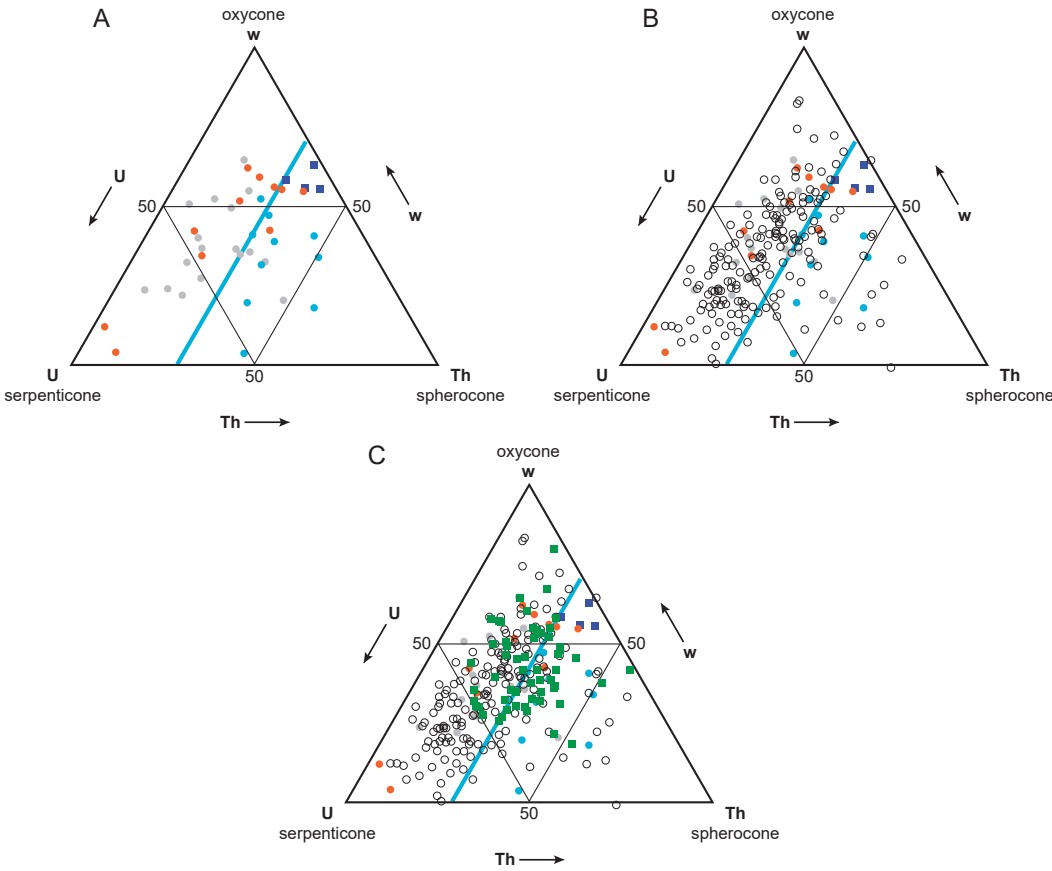

**Figure 5  Buoyant vs. non-buoyant cephalopods in Westermann Morphospace.** Note that plots are inverted relative to *Ritterbush & Bottjer (2012)* original ternary diagrams; this orientation is more standard for ternary diagrams. (A) *Reyment*'s *(1973)* taxa (circles) and modern nautilids (dark blue squares) in Westermann Morphospace. Modern nautilids include *Nautilus belauensis Saunders, 1981*, *N. macromphalus Sowerby, 1848*, *N. pompilius Linnaeus, 1758*, and *Allonautilus scrobiculatus Ward & Saunders, 1997*. Color coding indicates predicted postmortem buoyancy: buoyant–blue, intermediate–gray, not buoyant–orange. Light blue line represents a thickness ratio of 30%, generally separating buoyant from non-buoyant shell forms. (B) Addition of ammonoid taxa reported in *Ritterbush & Bottjer (2012*; open circles). Most ammonoids fall within the non-buoyant postmortem field of morphospace. (C) Addition of Late Cenomanian and Early Turonian cephalopod genera (green squares). These taxa show a mix of shell forms predicted to be buoyant and non-buoyant postmortem.

## Analysis of relationship between postmortem buoyancy and geographic range

Two approaches were used to assess the prediction that taxa with more buoyant postmortem shell forms would be more likely to drift and therefore have larger geographic ranges. First, the parameter *Th* was determined to be predictive of whether a taxon was likely to be buoyant or non-buoyant postmortem (see Results below, Fig. 5). Correlations between *Th* and geographic range were calculated separately for Late Cenomanian and Early Turonian genera and tested for significance (Fig. 6). To check for the influence of taxa with small geographic ranges, which could potentially represent sampling or collecting failures,

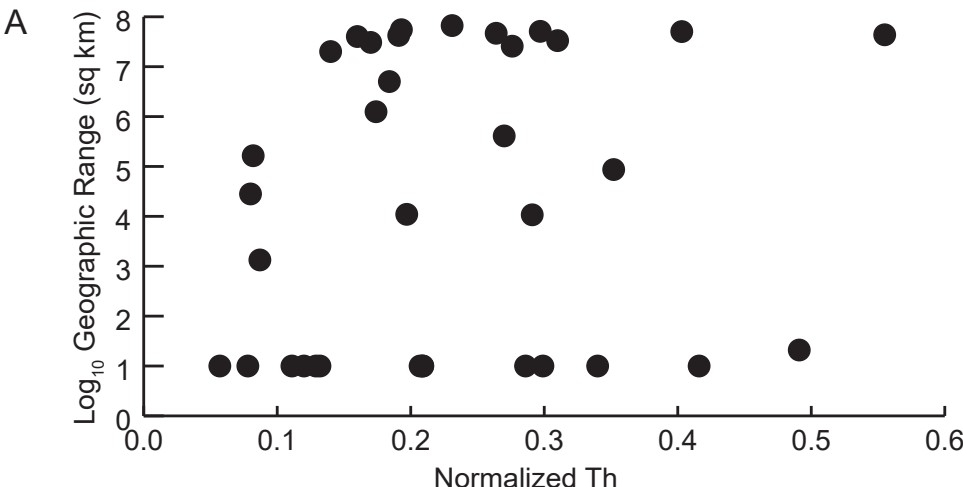

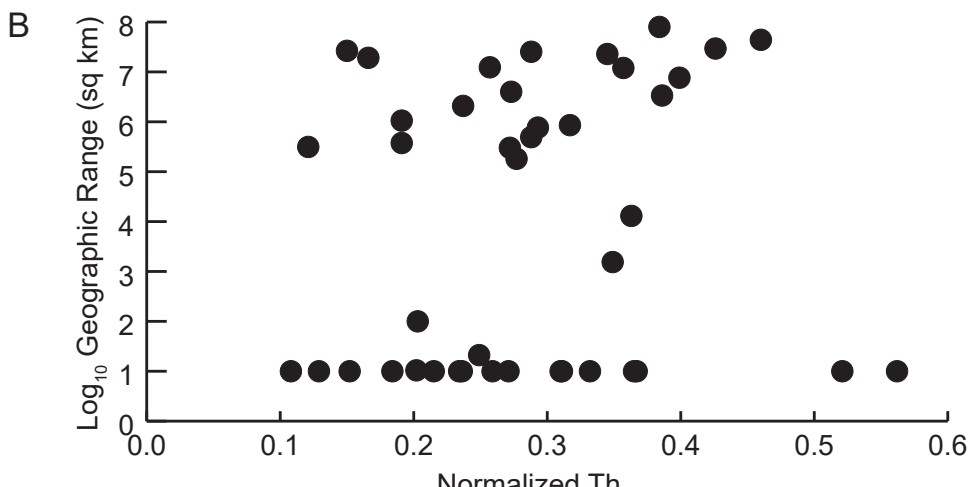

**Figure 6 Geographic range vs. shell shape.** Scatterplots of geographic range (expressed as the $\log_{10}$ of the area in square kilometers spanned by each genus' occurrences) versus normalized thickness ratio ($Th$) (reflecting degree of shell compression for that genus). (A) Early Turonian genera. Correlation coefficient $r = 0.161$ ($p = 0.355$). (B) Late Cenomanian genera. Correlation coefficient $r = 0.117$ ($p = 0.459$).

correlations were calculated for the full datasets and after excluding taxa with smaller ranges ($<1,000$ sq km). It should be noted that many of these taxa with small geographic ranges come from well-sampled regions (such as the Western Interior Seaway of North America and the European Platform) and therefore are more likely to reflect truly restricted ranges than sampling bias; in any case, these narrow ranges do not support the notion of extensive postmortem transport. Second, the distributions of geographic ranges for buoyant vs. non-buoyant genera were visualized and Mann–Whitney $U$-tests used to test for significant differences in the medians of these distributions (Fig. 7). All statistical tests were conducted in PAST 3.15 (*Hammer, Harper & Ryan, 2001*).

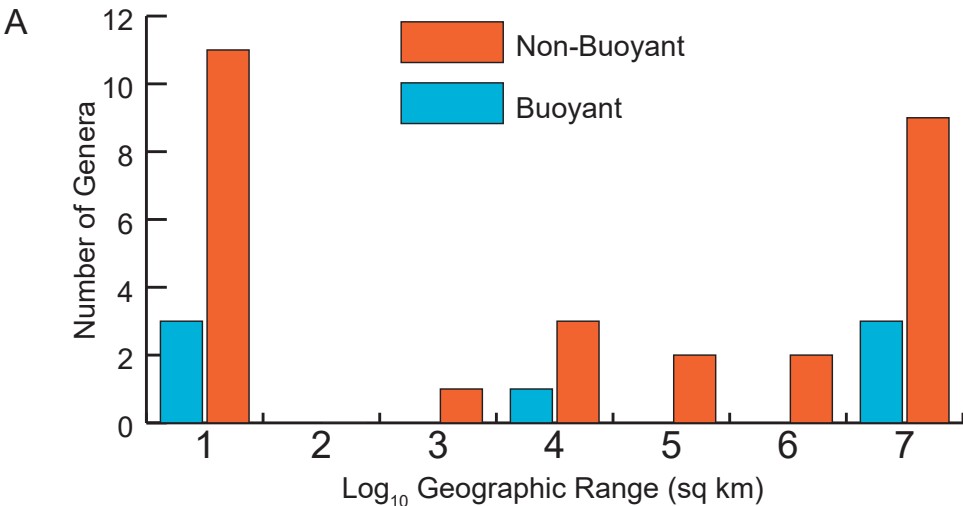

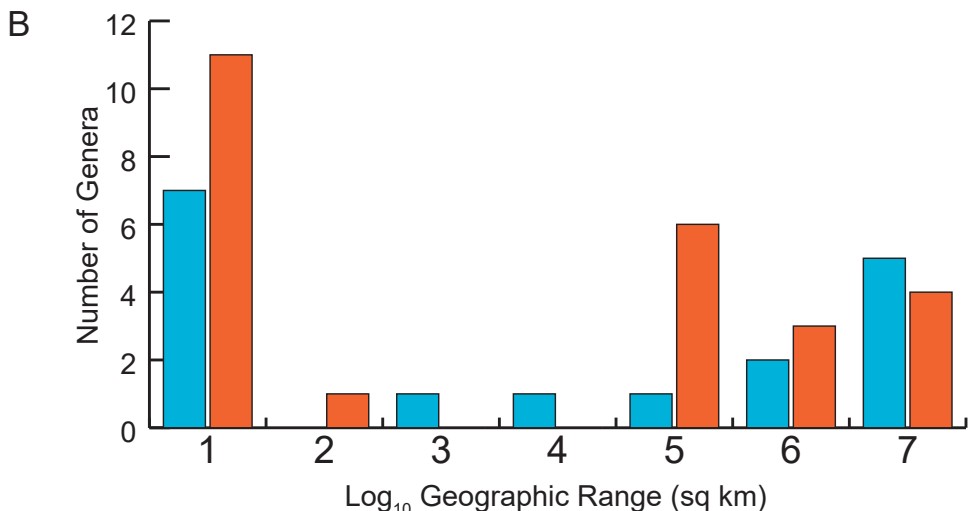

**Figure 7 Distributions of geographic ranges for taxa with buoyant vs. non-buoyant postmortem shell forms.** (A) Early Turonian genera. (B) Late Cenomanian genera. Distributions are not significantly different, implying that taxa with buoyant shell forms more likely to drift postmortem do not, in fact, show larger geographic ranges.

One complicating factor in assessing the causes of geographic range size is the potential swimming ability of the taxa investigated (*Ritterbush et al., 2014*). In *Ritterbush & Bottjer (2012)* construction of Westermann Morphospace, they mapped different interpreted modes of life onto their ternary diagram, with good nektonic swimmers represented in the oxyconic portion of the morphospace, vertical migrants with poor horizontal swimming ability in the spherocone region, and planktonic drifters in the serpenticone region. Demersal groups fell in a curved band separating the serpenticone corner from the rest of the morphospace. In order to control for the possible confounding relationship of

geographic range size with mode of life, Late Cenomanian and Early Turonian genera were assigned to one of three categories (nektonic, demersal, or vertical migrant) based on their position within Westermann Morphospace. No genera fell within the planktonic region; 19 genera falling near the center of this morphospace were excluded as their mode of life could not be unambiguously assigned. The non-parametric Kruskal–Wallis test was used to determine whether median geographic range was significantly different among modes of life.

### Geospatial analysis of modern nautilid occurrences
ArcGIS 10.3.1 (Esri, Redlands, CA, USA) was used to create maps of modern nautilid occurrences and to calculate convex hulls spanning the occurrences for each species.

## RESULTS
### Westermann Morphospace
The Mesozoic ammonoids whose buoyancy *Reyment (1973)* investigated plot separately in Westermann morphospace based on their postmortem buoyancy potential (Fig. 5A). The least buoyant taxa (orange circles) and most buoyant taxa (blue circles) mostly fall into separate fields, divided by the line corresponding to a normalized thickness ratio *Th* (100 × whorl width/shell diameter) of about 30%. Just two buoyant species fell in the non-buoyant field, and three non-buoyant species fell in the buoyant field. Fourteen of the 17 (82%) ammonoid taxa that *Reyment (1973)* identified as having an intermediate buoyancy (gray circles) fall in the non-buoyant area. It should be noted that the majority (66%) of Reyment's ammonoid taxa fall within the non-buoyant area. In contrast, the four living nautilid species (*Nautilus belauensis*, *N. macromphalus*, *N. pompilius*, and *Allonautilus scrobiculatus*; dark blue squares), which are known to experience postmortem transport, plot adjacent to the buoyant ammonoids, but with a higher value of *w*.

The majority (81%) of the 177 ammonoid taxa used in the original Westermann morphospace paper (*Ritterbush & Bottjer, 2012*), which represent a broad range of groups and time periods, fall within the non-buoyant field (Fig. 5B, open circles). This result suggests that most ammonoid morphotypes would have had a relatively low potential for postmortem flotation. Only ammonoid taxa with relatively wide, inflated shells would have a high potential for experiencing postmortem transport.

Ammonoid genera (plus the nautilid *Eutrephoceras*) from the Cenomanian-Turonian boundary interval plot in both the non-buoyant and buoyant fields (Fig. 5C, green squares). These results lead to the prediction that genera falling in the buoyant field (that is, with a normalized thickness ratio greater than 30%) may show artificially inflated geographic range sizes relative to those contemporaneous taxa with lower postmortem buoyancy, since more buoyant taxa would have been more likely to be transported by surface currents.

To confirm that buoyant vs. non-buoyant ammonoids differ in shell form, two other approaches to visualizing morphospace were investigated, PCA and CVA. Detailed results of these analyses are provided in Supplemental Information 1. These morphospaces also showed good separation of buoyant and non-buoyant shell forms, with buoyant shells

**Table 1 Comparisons of median geographic range sizes for Late Cenomanian and Early Turonian cephalopod genera.** Shells that are buoyant versus not buoyant postmortem are defined as having a normalized thickness ratio of more than versus less than 30% (see Fig. 4).

| | Median $\log_{10}$ geographic range (sq km) | Count | p (same median) from Mann–Whitney $U$-test |
|---|---|---|---|
| Late Cenomanian-Buoyant | 4.11 | 17 | 0.589 |
| Late Cenomanian-Not Buoyant | 5.26 | 25 | |
| Early Turonian-Buoyant | 4.94 | 7 | 0.688 |
| Early Turonian-Not Buoyant | 4.24 | 28 | |

falling in the PC or CV space associated with high loadings for whorl breadth $b$, thereby confirming the pattern observed in Westermann Morphospace.

## Relationship between predicted postmortem buoyancy and geographic range

There is no significant relationship between shell shape and geographic range. Correlations between the normalized thickness ratio and geographic range sizes of Cenomanian-Turonian cephalopods are weak: Early Turonian correlation coefficient $r = 0.161$ ($p = 0.355$) (Fig. 6A); Late Cenomanian correlation coefficient $r = 0.117$ ($p = 0.459$) (Fig. 6B). Excluding genera with geographic range sizes less than 1,000 sq km ($\log_{10} < 3$) did not result in significant correlations (Early Turonian $r = 0.399$, $p = 0.073$; Late Cenomanian $r = 0.140$, $p = 0.525$). Hence, the geographic range size of these cephalopod genera is not predictable based on shell form or the postmortem buoyancy inferred from that form, suggesting that postmortem transport was not an important factor in controlling observed geographic range.

In addition, no significant difference exists in the distributions of geographic range sizes for cephalopod shells predicted to be postmortem drifters vs. non-drifters. Genera with shell shapes associated with increased postmortem buoyancy (Fig. 7, blue histograms) do not show larger geographic ranges than less buoyant genera (Fig. 7, orange histograms) in the Late Cenomanian or Early Turonian. Rather, both groups of genera show $U$-shaped distributions; numerous genera had small geographic ranges regardless of their postmortem transport potential. Mann–Whitney $U$-tests reveal no significant differences in the medians of these distributions (Table 1). It seems clear that even if a cephalopod taxon has a shell shape predicted to be more buoyant postmortem, that potential did not in practice result in more postmortem transport and an inflated geographic range.

Further, no significant difference in median geographic range was found when grouping these Cenomanian and Turonian ammonoid genera based on their mode of life as predicted by their position in Westermann Morphospace (Table 2). Indeed, the genera with characteristics one might predict would result in larger observed geographic ranges, such as active nektonic swimmers, had smaller median geographic ranges than vertical migrants; perhaps these vertical migrants had less control over their distributions than actively swimming forms.

**Table 2  Comparisons of median geographic range sizes for Late Cenomanian and Early Turonian cephalopod genera by mode of life.** Mode of life is based on position within Westermann Morphospace, following *Ritterbush & Bottjer (2012)*. No vertical migrants were predicted to be non-buoyant postmortem and no nektonic forms were predicted to be buoyant postmortem, based on their normalized thickness ratios (see Fig. 5).

| | Median $\log_{10}$ geographic range (sq km) | Count | p(same median) from Kruskal–Wallis test |
|---|---|---|---|
| Nektonic | 1.32 | 17 | |
| Demersal | 3.61 | 32 | 0.323 |
| Vertical migrant | 7.47 | 9 | |

## Modern nautilid geographic distributions

It is interesting to note that Reyment's own buoyancy experiments (*Reyment, 1973*) showed that fossil cephalopods had a range of postmortem transport potentials, contradicting his persistent claim that most cephalopods experienced extensive transport (*Reyment, 2008*). The underlying rationale for Reyment's insistence that postmortem drift in shelled cephalopods is the norm was his claim that modern nautilids are known to frequently experience postmortem drift across long distances. This claim is based on observed occurrences of living nautilid specimens and beached shells interpreted as having drifted away from living populations as reported in *Toriyama et al. (1965)* and *House (1987)*. As noted in the Introduction, while drift has been observed, actual evidence for long-distance transport of nautilid shells is actually relatively rare (*Wani et al., 2005*).

Living populations of modern nautilids are known from the western equatorial Pacific, northern Australia, New Guinea, Indonesia, Palau, and the Philippines (Fig. 8A, orange circles). Drift shells, on the other hand, have been found over a much wider area, including further north in the Pacific, around southern Australia, and across the Indian Ocean (Fig. 8A, brown circles). Pooling all nautilid species together makes it appear that nautilids are prone to extensive postmortem transport.

However, investigating the geospatial distributions of individual nautilid species reveals a more complex pattern. Four of the five generally recognized nautilid (morpho)species (*N. belauensis*, *N. macromphalus*, *N. stenomphalus Sowerby, 1848*, and *Allonautilus scrobiculatus*; *Barord et al., 2014*) have relatively small living geographic ranges (Fig. 8B, Table 3). Drift shells of these species (darker circles) are rare and remain relatively close to the living population from which they are derived. For example, the one reported location of drifted *N. stenomphalus* falls within the known living range of the species. *N. belauensis* is known only from Palau, while drift shells have been recovered about 1,100 km away in Mindanao, Philippines (*House, 1987*). *N. macromphalus* lives in the area of New Caledonia, while drift shells have been found 2,100 km away in southeast Australia (*House, 1987*). While these distances are not trivial, they also do not extend the known range of these species to new ocean basins.

The exception to this pattern is *N. pompilius*, which has a much larger living range than the other nautilid species as well as drifted shells observed across a large part of the Indo-Pacific (Table 3, Fig. 8B, green circles). *House (1987)* and *Reyment (2008)* claimed that
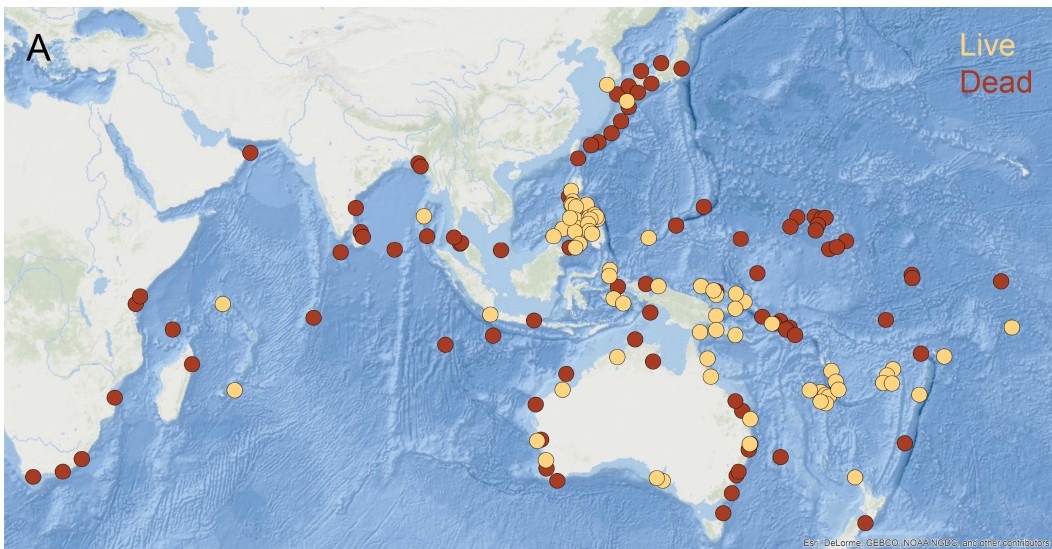

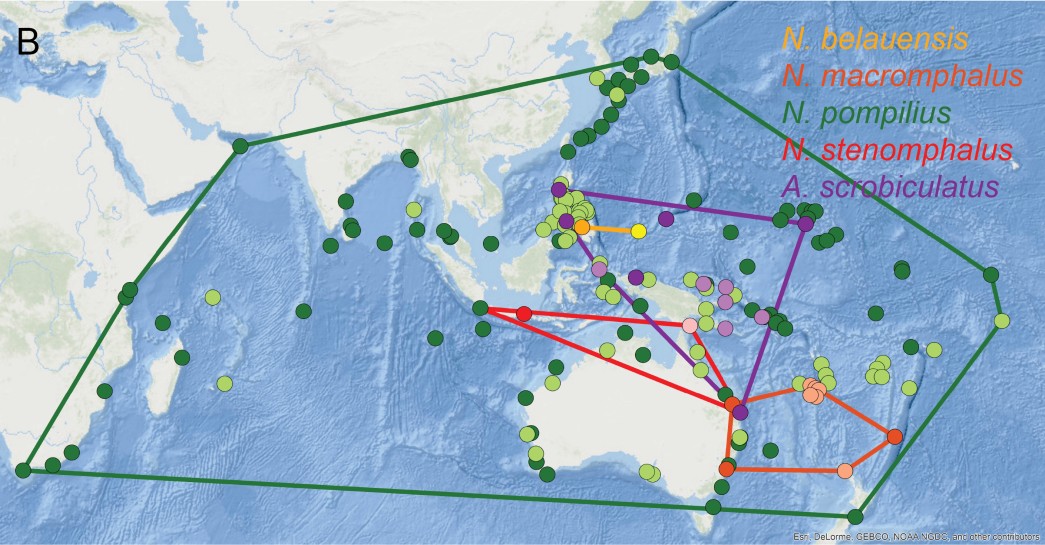

**Figure 8  Geographic distributions of modern nautilid species.** (A) All modern nautilid occurrences, including live specimens (orange) and dead shells (brown). Drifted shells span a larger area than known living occurrences. (B) Occurrences color-coded by species (live in lighter color, dead in darker color), with convex hulls enclosing all occurrences for each species. Most nautilid species do not have large geographic areas, even including shells that have drifted postmortem. The exception is *N. pompilius*, which has the largest range of drifted shells. The abundance of drifted shells in eastern Africa suggests the possibility of cryptic living populations of *N. pompilius* in the western Indian Ocean. Base map of Indo-Pacific region from *Esri (2016)*. Locality data from *Toriyama et al. (1965)* and *House (1987)*.

drifted shells in the western Indian Ocean must have derived from living populations in the Philippines, implying travel of dead shells over distances of over 9,000 km. It is not clear, however, why *N. pompilius* would be so much more capable of long-distance postmortem transport compared to the other living nautilid species, which are quite similar in their habitats, shell forms, and postmortem buoyancy and which can co-occur with *N. pompilius*
**Table 3  Geographic range sizes of modern nautilids.** Range sizes are represented as the area in square kilometers of convex hulls enclosing nautilid occurrences (Fig. 8B).

| Species | Living geographic range (sq km) | Living geographic range plus drift shells (sq km) |
|---|---|---|
| *Allonautilus scrobiculatus* | 1,458,480 | 11,808,955 |
| *Nautilus belauensis* | 500 | 2,200 |
| *N. macromphalus* | 185,977 | 4,369,601 |
| *N. stenomphalus* | 3,713,991 | 3,713,991 |
| *N. pompilius* | 76,428,763 | 131,322,000 |

and therefore experience the same oceanographic conditions and currents (*Saunders, 1987*; *Saunders & Ward, 1987*).

## DISCUSSION

### Postmortem drift in fossil cephalopods

The results presented here support the claim that the majority of ammonoid taxa (including both Paleozoic and Mesozoic groups) would have been considerably less likely to experience postmortem transport than modern nautilids. Only the subset of ammonoid taxa with relatively inflated shells would experience sufficient postmortem buoyancy to float for any length of time, a prerequisite for extensive transport away from their living habitat. Further, this conclusion does not consider that other factors, such as the depth at which the animal died, could further diminish the likelihood of postmortem drift, as animals in deeper water are more likely to become waterlogged and sink more quickly (*Chamberlain Jr, Ward & Weaver, 1981*; *Maeda & Seilacher, 1996*). Based on evidence from Late Cretaceous ammonoids, genera with shell forms conducive to postmortem floating did not have larger geographic ranges (Fig. 6), indicating that even those fossil cephalopods with a greater chance of postmortem transport did not likely actually experience extensive movement away from their living habitat. It should be noted that such shell forms are associated with a vertical migrant life mode (*Ritterbush & Bottjer, 2012*). The lack of significantly larger geographic ranges in such taxa indicates that neither transport during life nor postmortem drift markedly increased their observed geographic distributions. These results are consistent with previous workers' arguments for the rarity of extensive postmortem transport of fossil cephalopods, based on taphonomic evidence (*Kennedy & Cobban, 1976*; *Hewitt, 1988*; *Maeda & Seilacher, 1996*; *Maeda, Mapes & Mapes, 2003*; *Wani, 2004*; *Wani et al., 2005*; *Mapes et al., 2010a*; *Lukeneder, 2015*) and buoyancy considerations (*Chamberlain Jr, Ward & Weaver, 1981*; *Hewitt & Westermann, 1996*; *Chirat, 2000*; *Wani & Gupta, 2015*). They also provide a measure of confidence that fossil locality data are a robust proxy for living biogeographic ranges.

### Implications for modern nautilid distributions

The notion that postmortem transport was common in fossil cephalopods ultimately derives from the claim that modern nautilids show frequent and extensive postmortem drift. However, drifted nautilid shells are actually fairly rare (*Wani et al., 2005*) and only one extant nautilid species, *N. pompilius*, shows widely distributed drift shells. The limited

 

geographic distributions of most nautilid species are consistent with evidence for genetic divergence of geographically separated populations (*Wray et al., 1995*; *Bonnaud, Ozouf-Costaz & Boucher-Rodoni, 2004*; *Sinclair et al., 2007*; *Sinclair et al., 2011*; *Bonacum et al., 2011*; *Williams et al., 2015*; *Vandepas et al., 2016*). The wider distribution of *N. pompilius* may be real or the result of pooling multiple genetically distinct populations into single named species. *N. pompilius* does appear to contain more genetic variation than other nautilid species, and may be paraphyletic, with *N. stenomphalus* and *N. belauensis* falling within *N. pompilius* sensu latu (*Bonacum et al., 2011*; *Vandepas et al., 2016*; *Combosch et al., 2017*; *Saunders, Greenfest-Allen & Ward, 2017*). Hence, the wide distribution of *N. pompilius* drift shells could actually be caused by the conflation of smaller drift ranges of separate populations.

The abundance of drifted *N. pompilius* shells in the western Indian Ocean is particularly remarkable, as these sites are quite distant from known living populations (Fig. 8B). Both *House (1987)* and *Reyment (2008)* argued that the *N. pompilius* shells that wash up in some numbers on beaches in Kenya and Mozambique must have drifted across the entire Indian Ocean from living populations in the Philippines. Under this view, floating shells would need to move from the Philippines into the eastern Indian Ocean basin, where they could then be picked up by the westward flowing North or South Equatorial surface currents. This mechanism could conceivably bring floating shells across the Indian Ocean to Africa, but would require the shells to remain relatively undamaged (so as to continue floating) for a journey of over 6,000 km and to avoid being caught up in the eastward flowing Equatorial Countercurrent or the Indian Ocean gyre lying south of the equatorial currents (*National Weather Service, 2018*). In addition to these obstacles, it is difficult to understand why this one species of nautilid would show such different postmortem behavior than its close relatives. As an alternative, it might be possible that these African drift shells are actually being sourced from living populations much closer, in the western Indian Ocean basin (*House, 1973*; *Wani et al., 2005*; *Matteucci, 2015*). Live *Nautilus* (presumed to be *N. pompilius*) have been reported from Madagascar (*Teichert, 1970*) and from Mauritius and the Seychelles (by Dr. Anna Bidder, cited in *Reyment, 1973*), and late Pleistocene fossils of *Nautilus* (likely *N. pompilius*) have been identified on the Bajuni Islands off the Somali coast (*Matteucci, 2015*). The distances from the isolated islands of the western Indian Ocean to the African mainland are more consistent with the distances traveled by dead shells of the other nautilid species. The possibility of living *N. pompilius* populations within the western Indian Ocean basin is therefore worthy of further investigation, especially in light of new conservation efforts for this iconic marine animal. In addition to a search for living populations in the waters surrounding Madagascar, Mauritius, the Comoros, the Seychelles, and in reef settings off the East African coast, further investigation of potential nautilid fossils from late Pleistocene deposits, like those described by *Matteucci (2015)* from Somalia, is warranted. Recovered drift shells of nautilids from East Africa should also be assessed for age (for instance, by radiocarbon or amino acid racemization techniques; *Tomašových et al., 2017*) and possible sequenceable organic material, which would allow the placement of these shells in the larger phylogenetic context of nautilids throughout the Indo-Pacific region.

Such new discoveries are not without precedent. A second species of coelacanth, *Latimeria menadoensis* (*Pouyaud et al., 1999*), was identified in Indonesia in 1997, nearly 10,000 km east across the Indian Ocean from the coelacanths' known range in east Africa (*Pouyaud et al., 1999*). Genetic data suggest these two species have been separated for tens of million years (*Holder et al., 1999*; *Inoue et al., 2005*). It seems clear that the marine biota of the Indian Ocean, especially deeper water species like the nautilus and the coelacanth, is still incompletely known.

## CONCLUSIONS

While postmortem transport of cephalopod shells in modern and fossil contexts is frequently assumed to have been both frequent and extensive, evidence supporting this view is lacking. Most fossil ammonoids had shell forms that resulted in low postmortem buoyancy; only highly inflated shells were likely to float for significant periods after death. An analysis of early Late Cretaceous (Cenomanian-Turonian) cephalopod genera shows that observed geographic range size was not related to likelihood of postmortem transport, indicating that these observed fossil ranges are adequate proxies for living geographic ranges. Most living nautilid species have relatively small geographic ranges with limited dispersal of drift shells. The exception, *Nautilus pompilius*, is widespread throughout the Indo-Pacific, with drift shells found apparently great distances from known living populations. However, drift shells found along the east African coast may in reality be derived from cryptic living populations among the isolated islands of the western Indian Ocean.

## ACKNOWLEDGEMENTS

The author would like to thank R Hoffmann, N Landman, B Linzmeier, R Mapes, and K Ritterbush for helpful discussions, and A Avruch, K De Baets, R Wani, R Lemanis, and an anonymous reviewer for very constructive reviews of the manuscript. This paper is dedicated to the memory of Richard Reyment, who passed away in 2016; his pioneering combination of experimental and quantitative analyses of ammonoid form and function have made work like this possible.

### Funding

The authors received no funding for this work.

### Competing Interests
The authors declare there are no competing interests.

### Author Contributions
- Margaret M. Yacobucci conceived and designed the experiments, performed the experiments, analyzed the data, contributed reagents/materials/analysis tools, prepared

figures and/or tables, authored or reviewed drafts of the paper, approved the final draft, single author paper.

## Data Availability

The raw data are provided in a Supplemental File.

## Supplemental Information

Supplemental information for this article can be found online at http://dx.doi.org/10.7717/peerj.5909#supplemental-information.

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
