# Peer review of "Postmortem transport in fossil and modern shelled cephalopods"

_PeerJ, doi:10.7717/peerj.5909_

## Round 0.1 · original submission · Minor Revisions

This is a very original and well-written contribution providing new data and analysis on postmortem drift of externally shelled cephalopods. I have little doubt it will become an important contribution to the field as attested by positive reviews and my own personal assesment. However, there are some crucial points I would still like to see addressed before publication:

The case for the increased interest in ammonoid biogeography and quantitative analysis: I agree that disentangling biogeographic distribution of ammonoids is of great interest, but I feel it is still too often done very qualitatively. I feel it would be more appropriate to cite the various references you cite in the context of biogeography for the interest and need for quantitative biogeographic analysis rather than for its application – in my perception only a few good quantitative analyses have been published (Brayard et al., 2007; Dera et al., 2011; Zacaï et al., 2016; Zacaï et al., 2017). A good recent example on the Cretaceous might be Rojas et al. (2017). I think the latter ones are worth citing in the context of interest in quantitative analysis of ammonoid biogeographic distribution.

Buoyant, non-buoyant and intermediate categories of Reyment: It should be more elaborately explained what Reyment actually did (see also reviewer 2). I agree with reviewer 3 that the categories of Reyment do not seem to be very well defined in your ternary diagram. Would it maybe be possible to do a discriminant analysis to figure out how his categories could be separated best and/or which areas are consistently separated best. This would be more similar to Reyment original definitions. I do agree that thickness ratio is also a fairly good predictor, but I would like to see both to analyze the categories. It would also be interesting to relate Reyment categories with modern empirical or theoretical calculations of buoyancy with an empty phragmocone (see reviewer 1), although an extensive treatment might be buoyant the scope of this paper (see also reviewer 3).

Trajectories after death: Various authors have argued for the influence of depth on post-mortem history. At least some authors have argued for sinking (instead of surfacing) if ammonoids died too deep (Maeda and Seilacher, 1996) which could of course influence post-mortem transport, too (see more extensive comments by reviewer 2).

Swimming capabilities versus drifting: I really like your approach of using Reyment predictions to infer buoyancy of ammonoids in the Westermann “morphospace” . However, it has been argued that some of these characters might also impact on swimming capabilities of these ammonoids and their distribution in the water column (Ritterbush and Bottjer, 2012; Ritterbush et al., 2014). I am only aware of Tanabe (1979) who discussed both as alternative models to explain ammonoid distribution. From this perspective, it might also be worth testing how additional categories based on hypothesized swimming capabilities (or their interactions) perform with respect to geographic distribution and analyze also the potential different response of groups which are poorly buoyant, but good swimmers and groups which might be buoyant, but not good swimmers. It could happen that poorly buoyant shells are counterbalanced by large migrations during life or vice versa that buoyancy after death in others counterbalances the limited geographic distribution during life. I understand that this might take out a bit of the power from the analysis due to smaller sample size but I feel it needed to strengthen your interpretation. One could potentially merge Turonian and Cenomanian datasets for this additional analysis. I feel it would make the manuscript even more interesting. Even if would plot your line over the traditional interpretations (Ritterbush and Bottjer, 2012) – although these are not without controversy, you could have 4 categories: plankton, nekton, vertical migrant or demersal – the last one is a tricky one as it overlaps with multiple fields. You could check if these categories explain better the differences in range within your dataset(s). Although I guess one could also make 3 broader ones: a buoyant, ok swimming spherocone field (although their vertical migrant properties could affect post-mortem floating as sinking or floating history is thought to be dependent on depth), a non-buoyant, better swimming one (oxycone) and a non-buoyant, less swimming one (serpenticonic).

Nautilus taxonomy: it is clear that the only feasible approach for your purposes (e.g., to identify isolated and compare with fossil shells) is a morphospecies concept, but it might be good to state this more clearly. From your table 2, I gather that you are using 5 accepted species of recent nautilids by most authors (e.g., http://www.marinespecies.org/aphia.php?p=taxdetails&id=153112). It might however be worth pointing out that there might actually more populations/species based on molecular approaches and that these not necessarily all correspond with the morphologically defined species (Bonacum et al., 2011; Combosch et al., 2017; Vandepas et al., 2016). Nautilus pompilius, in particular, has been variously considered as paraphyletic lineage (Bonacum et al., 2011) or a very variable species (Vandepas et al., 2016) based on MtDNA. This could also affect its successful identification in isolated shell samples and contribute to overestimating its geographic range. The situation might even be more complex using genomes as suggested by (Combosch et al., 2017) with several “cryptic” species, but little to no genetic substructure in the Indo-Pacific – the latter could again be in line with the widespread “N. pompilius”. I think it would be appropriate to at least discuss how these results might affect your results.

Existence of unverified modern (or recently extirpated) living populations in the western Indian ocean: I have no issue with this suggestion, but in theory this could be tested more extensively not only by looking for recent populations, but also dating drifted shells attributable to these regions. It would be appropriate to discuss how one could test this hypothesis in the future. Amino Acid Racemization could for example be used in the latter case to verify rather recently extirpated living populations.

Figure 2: I think this figure should be in, but it not clear to me how different categories of Reyment are defined within this figure. Would it not be possible to provide a simplified picture.

Please also address these additional points as well as those of the reviewers, in addition to these points:

Line 18: it might be interesting to see how swimming abilities might confound the signal – as there are better swimmers with a non-buoyant shell and poorer swimmers with a buoyant shell.
Line 59: I agree that the lack of soft parts have often been used as an argument, but the lack of soft-tissues could also have other reasons (Ritterbush et al., 2014) – including physiological ones (Clements et al., 2017).
Line 61: I guess it might be worthwhile mentioning here or further down the line that “older” shells could potentially be dated using amino acid racemization (Tomašových et al., 2017; Tomasovych et al., 2016). Furthermore, is there a possibility that such finds could potentially be attributable to transport by “historical” ship(wreck)s?
Line 82: I think time Tanabe (1979) might also be citable in this context.
Line 85: The following references also discuss that most shells in death assemblages are at most a couple 1000 years old, few older ones are markedly degraded (Tomašových et al., 2017; Tomasovych et al., 2016).
Line 108: it would sound better if you would write nautilids as well as ammonoids
Line 123: I guess you mean living ones as many additional analyses are known to trying to infer buoyancy of shells.
Line 131: “float until” – it has been argued that shells might sink if cephalopod did in greater depth (Maeda and Seilacher, 1996).
Line 237, 238: it is not really what a, a’ and b stand for; please elaborate. The name given by Ritterbush and Bottjer is a bit confusing as it is not a true morphospace, but rather a projection of one.
Line 300: one could also wonder how the distribution of epicontinental seas plays into this issue. They are quite extensive in Cretaceous – maybe less so today?
Line 284-285: yes, but I am wondering how swimming abilities during life might influence their distribution. Could it be that you are mixing less buoyant, more swimming and more buoyant, less swimming forms which could partially confound this signal?
Line 327: I guess it would be better to speak of morphospecies or at least refer to the 5 accepted recent species by most authors: http://www.marinespecies.org/aphia.php?p=taxdetails&id=153112).
Line 337: This could however be an artefact as morphologically defined species might partially represent cryptic paraphyletic groups
Line 342-344: I wonder how ocean currents could potentially influence this pattern
Line 367: yes, but it is not clear if pompilius is actually one species (Bonacum et al., 2011) which might affect this interpretation
Line 382: fossils of Nautilus – which species and what is their more exact age? Furthermore, isolated Holocene finds could potentially be dated with amino acid racemization.
Figure 7: I guess molecular results could be used to partially revise these range for some taxa into particular regions as most molecular studies show quite some biogeographic structure with the exception of the Indopacific (Combosch et al., 2017).
Table 1: it would be interesting to see what happens if you have 3 categories based on swimming capabilities versus buoyancy categories
Table 2: it might help to work with proportional ranges (log ratio) to compare ranges. The range of drifted shells of Allonautilus is almost an order of ten bigger and that of N. macromphalus almost 20 bigger, while that of N. pompilius is only double as big, etc.


Suggested references
Bonacum, J., Landman, N. H., Mapes, R. H., White, M. M., White, A.-J., and Irlam, J., 2011, Evolutionary Radiation of Present-Day Nautilus and Allonautilus: American Malacological Bulletin, v. 29, no. 1-2, p. 77-93.
Brayard, A., Escarguel, G., and Bucher, H., 2007, The biogeography of Early Triassic ammonoid faunas: Clusters, gradients, and networks: Geobios, v. 40, no. 6, p. 749-765.
Clements, T., Colleary, C., De Baets, K., and Vinther, J., 2017, Buoyancy mechanisms limit preservation of coleoid cephalopod soft tissues in Mesozoic Lagerstätten: Palaeontology, v. 60, no. 1, p. 1-14.
Combosch, D. J., Lemer, S., Ward, P. D., Landman, N. H., and Giribet, G., 2017, Genomic signatures of evolution in Nautilus—An endangered living fossil: Molecular Ecology, v. 26, no. 21, p. 5923-5938.
Dera, G., Neige, P., Dommergues, J.-L., and Brayard, A., 2011, Ammonite paleobiogeography during the Pliensbachian–Toarcian crisis (Early Jurassic) reflecting paleoclimate, eustasy, and extinctions: Global and Planetary Change, v. 78, no. 3–4, p. 92-105.
Maeda, H., and Seilacher, A., 1996, Ammonoid Taphonomy, in Landman, N. H., Tanabe, K., and Davis, R. A., eds., Ammonoid Paleobiology: New York Plenum Press, p. 543-578.
Ritterbush, K. A., and Bottjer, D. J., 2012, Westermann Morphospace displays ammonoid shell shape and hypothetical paleoecology: Paleobiology, v. 38, no. 3, p. 424-446.
Ritterbush, K. A., Hoffmann, R., Lukeneder, A., and De Baets, K., 2014, Pelagic palaeoecology: the importance of recent constraints on ammonoid palaeobiology and life history: Journal of Zoology, v. 292, no. 4, p. 229-241.
Rojas, A., Patarroyo, P., Mao, L., Bengtson, P., and Kowalewski, M., 2017, Global biogeography of Albian ammonoids: A network-based approach: Geology, v. 45, no. 7, p. 659-662.
Tanabe, K., 1979, Palaeoecological analysis of ammonoid assemblages in the Turonian Scaphites facies of Hokkaido, Japan: Palaeontology, v. 22, no. 3, p. 609-630.
Tomašových, A., Schlögl, J., Biroň, A., Hudáčková, N., and Mikuš, T., 2017, Taphonomic Clock and Bathymetric Dependence of Cephalopod Preservation in bathyal, sediment-starved environments: PALAIOS, v. 32, no. 3, p. 135-152.
Tomasovych, A., Schlogl, J., Kaufman, D. S., and Hudackova, N., 2016, Temporal and bathymetric resolution of nautiloid death assemblages in stratigraphically condensed oozes (New Caledonia): Terra Nova, v. 28, no. 4, p. 271-278.
Vandepas, L. E., Dooley, F. D., Barord, G. J., Swalla, B. J., and Ward, P. D., 2016, A revisited phylogeography of Nautilus pompilius: Ecology and Evolution, v. 6, no. 14, p. 4924-4935.
Zacaï, A., Brayard, A., Dommergues, J.-L., Meister, C., Escarguel, G., Laffont, R., Vrielynck, B., and Fara, E., 2016, Gauging scale effects and biogeographical signals in similarity distance decay analyses: an Early Jurassic ammonite case study: Palaeontology, v. 59, no. 5, p. 671-687.
Zacaï, A., Fara, E., Brayard, A., Laffont, R., Dommergues, J.-L., and Meister, C., 2017, Phylogenetic conservatism of species range size is the combined outcome of phylogeny and environmental stability: Journal of Biogeography, v. 44, no. 11, p. 2451-2462.

Reviewer 1 ·

Basic reporting

English is good, the ms is well-structured, with supporting references. The figures are generally good but some improvement can be made. Figure 1might not be so informative for people in this field but it may be good for the general public. Figure 2 is a copy of a figure in Reyment (1973), but what he exactly did (what kinds of conch parameters he measured and so on) is not clear. This is important because the argument of this paper is mostly based on the experiments in Reyment (1973). For Figure 3, I would suggest adding some explanations of what are w, U and Th, (like w = a/a’ and etc..). The contrast between the orange and brown circles in Figure 7A is not clear. The samegoes for green circles in Figure 7B. I would suggest making a clearer contrast.

Experimental design

The topic of ammonoid and Nautilus postmortem transport is of great interest not only for cephalopod paleontologists but also for a broader range of palaeontologists and biologists.
The research question is clearly stated in a good context.
The methods used are clearly explained to the extent that this study is reproducible.

Validity of the findings

The author’s argument is largely based on the experiments carried out by Reyment (1973), in which he found that the various morphologies of ammonoids correspond to differing buoyancy. But I am wondering about the validity of his study. There are many newer studies which discuss ammonite buoyancy, as the author mentions in the text. I would suggest checking the validity of Reyment (1973) and discussing it thoroughly in the ms. I quickly looked at some data published a couple of years ago by Naglik et al. (2015; Lethaia) and Tajika et al. (2015; Historical Biology), and then calculated masses of the ammonoids without soft tissue (Fidelites, Diallagites, Goniatites and Normannites) and water displaced by the shells. It looks like all the ammonoid shells are positively buoyant in the sea, when phragmocone is empty. When these ammonoids are plotted in Westermann Morphospace, however, it appears that Fidelites and Diallagites are plotted in the “non-buoyant field”, whereas Goniatites and Normannites in the “buoyanct field”. This emphasizes the importance of discussing Reyment’s study (whether it is consistent to newer studies and etc.) more comprehensively. Since no empirical 3D models are available other than the above mentioned ammonoids, the author would be able to apply theoretical models of ammonites to test buoyancy of various ammonoid shell shapes to discuss this. Oxyconic and serpenticonic ammonoids would be very important here. If this kind of work was formerly carried out by some others, it should be discussed here as well.

The conclusion is clearly stated, which is also consistent to the results. The initial questions are answered in the Conclusion.

Additional comments

Overall, I think this paper is good. The postmortem transport potential of cephalopod shells is a key issue for various important paleontological studies. This well-written paper will surely contribute to better understanding of this topic. Although I would recommend this paper for publication, I also would suggest revising the ms before acceptance. Some more comments are as follows.

Line 43: I think a couple of references should be added here.
Line 129–130: Are there any previous studies which observed this?
Line 136–140: If it takes soft tissue a few to several days to get detached from the shell, I would assume that there is also transport to some degree during the time, which is not discussed in the text. Does it affect also the transport potential? This must be related to buoyancy of living animals with soft tissue.
Line 164–166: What happens if you use other morphospace like PCA? Reyment’s and Ritterbush and Bottjer’s species can be also plotted into a non-Westermann morphospace. Is there a specific reason to use Westermann Morphospace for this study? I think you can easily draw a buoyant/ non-buoyant line in PCA as well. I am saying this because I am slightly afraid about some possible modification of original data when you translate the morphometry datasets to Westerman Morphospace parameters (w, Th and U) due to the scaling and normalizing. It may be good to compare results of different morphospaces.
Line 191–193: The author states that original measurements in Reyment (1973) were not available, and thus congeneric species had to be used in some cases. This statement sounds like Reyment’s study is actually not reproducible. I would strongly suggest using at least the same species as Reyment used because morphology within a genus (even within a species) often considerably varies. In particular, whorl width (Th in Westerman Morphospace) should be an important diagnostic characters in ammonoids in general. Finding the same species and taking measurements should not be a lot of work.
Line 195–196: If readers do not have Ritterbush and Bottjer (2012) with them, it is hard to imagine what these parameters are. I would suggest adding a small figure with an ammonite cross section together with these parameters.
Line 206–208: There are no explanations for U, Th, and w here, even though these abbreviations appear for the first time here in the ms. The author explains these in Line 232–248. I would suggest adding a short phrase like “for definition of these parameters see Westermann Morphospace and postmortem buoyancy below”.
Line 232: Same as above. The parameters should be visualized.
Line 269–280: Does the buoyancy increase as Th increases? As stated in Validity of the findings, there appears to be no difference in buoyancy between 4 empirical ammonite models in Naglik et al. (2015) and Tajika et al. (2015). This should be discussed in the method or introduction section.
Line 275: What is the definition of intermediate buoyancy? Is it neutral buoyancy?
Line 281–286: Is there any correlation between the size (conch diameter) of ammonites and position in the field? The size has been discussed in previous studies. I am wondering if the author had a look at it. In fact, there is no information on ontogenetic stage and size of ammonites used for this study, which should be also important.
Line 322–324: I think it is better to make a stronger contrast between the two different circles in Figure 7A. The current ones are not easy to differentiate.
Line 330: What are data sources for the geographic ranges? References would be needed.
Line 377: Maybe N. pompilius is much more abundant than the other species? Just a thought.
Line 386: Looking at Figure 7B, there were also reports on living Nautilus in southern Japan and off Burma. I wonder how the author interprets this. Is it also worth investigating these places? Did these individuals just get lost from their original populations?

·

Basic reporting

I feel that this manuscript is well designed to PeerJ. English is very clear (at least I feel so), because this manuscript is easily understandable even for non-native speaker (=me). The manuscript has sufficient introduction and background to demonstrate. Figures are beautiful, which is relevant to the content of the manuscript. The raw data is shown in figures and supplementary file. Therefore, in this point of view, the manuscript totally follows the standards of PeerJ.

Experimental design

I feel that the experimental design of this manuscript is well organized. The investigation of the manuscript was conducted rigorously with a high technical standard. Therefore, in this point of view, the manuscript totally follows the standards of PeerJ.

Validity of the findings

I found one point that should be revised, to increase the robustness.
(1) How to classify buoyant or non-buoyant shells.
How to classify buoyant or non-buoyant species is crucial to this study. This totally depends on the assessments of Reyment (1973). I feel that the Reyment’s assessment are better to be explained more in detailed, which increase the robustness of this interesting study. In relation to this, I have some questions.
(a) What are the definitions of “buoyant”, “intermediate”, and “non-buoyant”? I cannot understand what these terms means and what are differences. Did shells of such “intermediate” species float, or sink down without surfacing or keep almost neutral buoyancy even after death? After surfacing, “buoyant” shells never sink down? If such shells sink down after some periods of floatation (is this “intermediate”?), how long such shells float and drift? Also, it is much better to explain the definition of “non-buoyant”. Did shells of such “non-buoyant” species sink down without surfacing? If shells of “non-buoyant” species sank down after a short interval of surfacing, this is not “non-buoyant”, which should be mentioned as “least-buoyant”. In line 270, the author mentioned “The lease buoyant taxa”, however in line 274, the author mentioned “non-buoyant”. I feel it is better to discriminate these terms, which mean different post-mortem history (”non” is not equivalent to “least”, at least I think so).
(b) Can we really recognize the taphonomic history (buoyant or non-buoyant) only from shell shapes? I actually have some doubt on the Reyment’s assessments. Even if shell shapes are similar, two species could live in different depth. If so, the taphonomic history of these species are expected not to be same, because the infilling rates of seawater into empty phragmocones after the removal of soft parts are expected to be different, which is related to what water depth the soft parts were removed. This is fully examined in Chamberlain et al. (1981), however, the Reyment’s assessments is in 1973, so that the assessments are probably not include such points of views. Therefore, I cannot understand why “buoyant” or “non-buoyant” can be recognized from shell shapes only.
(c) The infilling rates of seawater into empty phragmocones after the removal of soft parts would also depend on the characteristics of siphuncle tubes. Did the Reyment’s assessments contain this point of view?
(d) If the Reyment's assessments suppose that shells are not to be flooded and therefore never sink down after surfacing, this scenario would be not close to the true, considering the results of Wani et al. (2005). The Reyment’s assessments is in 1973, so that the assessments are probably not include this point. Even so, how can we recognize “buoyant” or “non-buoyant” shells?

(2) Several minor suggestions
I added comments of pdf files, which are minor suggestions. Some of my comments might be trivial or due to my misunderstanding, if so, please ignore it.

Additional comments

This manuscript is of great interest for many paleontologists, especially cephalopod workers, because it provides an interesting information that (1) only highly inflated shells were likely to float for significant periods after death, and (2) the observed fossil geographic range are adequate proxies for living geographic ranges. However, I found some points that should be revised, to increase the robustness. Therefore, I would like to strongly recommend accepting this interesting manuscript for the publication in PeerJ, with minor revision.

·

Basic reporting

The author has written a particularly interesting contribution about the contentious issue of port-mortem drift influence in cephalopod biogeography. Overall the contribution is well written and provides a good argument against drift having a heavy influence on biogeographic distribution of ammonites. Moreover, the suggestion of cryptic Nautilus populations is important for potential conservation efforts.

Experimental design

The research certainly fits within the scope of PeerJ. The goals are well stated and the data provided is well documented and annotated. Some comments about the approach are included in the general comments section.

Validity of the findings

I have some concern about the validity of the underlying buoyancy evaluation but validating this data is beyond the scope of the submitted paper. However, the distribution of buoyant shells in the morphospace, as noted in the general comments, doesn't seem to fit the described results.

Additional comments

Pg. 9, line 94 – The author brings up the issue of septal neck length on potential drift behaviour. While I understand this parameter couldn’t be included in the morphological analysis it does bring up the question of Spirula. Spirula has septal necks extending for most of the chamber and the shells do have evidence of port-mortem drifting. The dynamics are of course somewhat different since the shell is internal, but the occurrence of encrusting organisms on the shell and final septum would suggest at least some drifting after soft-tissue detachment. I’m curious how Spirula fits into this picture.

Pg. 14, line 190. Since you had to re-measure Reyment’s data, it perhaps would be nice to try to recreate the PCA of Reyment with your data in order to assess potential differences between your data and his and to assess the reproducibility of Reyment’s data.
This also begs the question, why did you do a Westermann morphospace at all? Why not just do a PCA analysis. I’m a bit unclear as to what benefit the Westermann morphospace has especially after doing a PCA with your data the results are fairly similar.

Pg. 14, line 197, 203-205. Considering that the only shell we have for a “control” for buoyancy assessment is nautilus and variation among extant forms is fairly small, how much do you really trust Reyment’s potentially reductive assessment of buoyancy?

Pg. 15, line 215. How exactly did you differentiate between artificially large geographic ranges and “true” geographic ranges for ammonoids?

Pg. 16, line 240. In the supplementary data, it would be nice to label the data from Ritterbush and Bottjer the same as the other data, i.e., if the U, Th, W, are raw/scaled or normalized.

Pg. 17, line 273. Here you state that the plotted data from Reyment only had two buoyant shells plotting in the non-buoyant field. But when I replotted the data I noticed a cluster of shells from this dataset that plot in the non-buoyant area, including Kamerunoceras, Ovaticeras, Eoderoceras, Amaltheus, among others. What is going on with these guys?
If post-mortem drift occurs infrequently, I would think that some groups would have occurrences that would cluster around their living occurrence area with maybe a handful of occurrences outside of this geographic range. Are there any ammonoid groups that show this kind of distribution? What parameters might we look for in judging a potential drift shell in the fossil record?

---

## Round 0.2 · Minor Revisions

Thank you addressing our comments and suggestions particularly the additional analysis on how geographic range relates with inferred swimming capabilities. Your paper is as good as accepted. I just had some additional minor points before publication:

1) Influence of oceanic currents. Although not explicitly mentioned, oceanic currents might also have an influence on where drifting shells might end up. As far as I can check, both drift from the Philippines, India and closer localities would be consistent with major current directions (although some might potentially drift longer if they are caught in loops near coast of India, etc.). A small mention on how current could affect distance and direction of drift would be appropriate.

2) Impact of taxa with short geographic ranges: When reading the manuscript in detail again, I noticed that on Figure 6, that a substantial part of your taxa have short geographic distributions (~ 10 sq km). These taxa show little relationship between geographic range and normalized Th, while the taxa with greater geographic range seem to show a slightly better relationship. On the one hand, it could indicate drifting did not play a large role at short distances, otherwise they would be more widely distributed. However, can you rule out certain preservational, collection or sampling biases are responsible for the short geographic distribution of these taxa. Even if there is a relationship, there could still exist a better one with other parameters.I would like you to briefly discuss how removing those taxa influences the patterns and how this could alter your interpretations. I guess it also relates to a more methodological/philosophical issue – should those taxa with short ranges be included or not. I feel a brief discussion on these aspects would be appropriate.

Please address the comments in the annotated pdf in addition to these two points.

Looking forward your revised manuscript.

---

## Round 0.3 · accepted · Accept

Thank you for addressing my final suggestions. Particularly, the definition of "drifted" shells and additional analyses excluding taxa with short geographic ranges make the manuscript even easier to follow. Looking forward to its publication.

#